# FeCAM: Exploiting the Heterogeneity of Class Distributions in Exemplar-Free Continual Learning

**Dipam Goswami**[1,2]   **Yuyang Liu**[3,4,5,†]   **Bartłomiej Twardowski** [1,2,6]   **Joost van de Weijer**[1,2]

[1]Department of Computer Science, Universitat Autònoma de Barcelona
[2]Computer Vision Center, Barcelona  [3]University of Chinese Academy of Sciences
[4]State Key Laboratory of Robotics, Shenyang Institute of Automation, Chinese Academy of Sciences
[5]Institutes for Robotics and Intelligent Manufacturing, Chinese Academy of Sciences  [6]IDEAS-NCBR
`{dgoswami, btwardowski, joost}@cvc.uab.es, sunshineliuyuyang@gmail.com`

## Abstract

Exemplar-free class-incremental learning (CIL) poses several challenges since it prohibits the rehearsal of data from previous tasks and thus suffers from catastrophic forgetting. Recent approaches to incrementally learning the classifier by freezing the feature extractor after the first task have gained much attention. In this paper, we explore prototypical networks for CIL, which generate new class prototypes using the frozen feature extractor and classify the features based on the Euclidean distance to the prototypes. In an analysis of the feature distributions of classes, we show that classification based on Euclidean metrics is successful for jointly trained features. However, when learning from non-stationary data, we observe that the Euclidean metric is suboptimal and that feature distributions are heterogeneous. To address this challenge, we revisit the anisotropic Mahalanobis distance for CIL. In addition, we empirically show that modeling the feature covariance relations is better than previous attempts at sampling features from normal distributions and training a linear classifier. Unlike existing methods, our approach generalizes to both many- and few-shot CIL settings, as well as to domain-incremental settings. Interestingly, without updating the backbone network, our method obtains state-of-the-art results on several standard continual learning benchmarks. Code is available at `https://github.com/dipamgoswami/FeCAM`.

## 1   Introduction

In Continual Learning (CL), the learner is expected to accumulate knowledge from the ever-changing stream of new tasks data. As a result, the model only has access to the data from the current task, making it susceptible to *catastrophic forgetting* of previously learned knowledge [46, 37]. This phenomenon has been extensively studied in the context of Class Incremental Learning (CIL) [36, 61, 9, 74], where the objective is to incrementally learn new classes and achieve the highest accuracy for all classes encountered so far in a task-agnostic way without knowing from which tasks the evaluated samples are [56]. While one of the simplest approaches to mitigate forgetting is storing exemplars of each class, it has limitations due to storage and privacy concerns, e.g., in medical images. Hence, the focus has shifted towards more challenging exemplar-free CIL methods [80, 63, 43, 41, 35, 2].

In exemplar-free CIL methods, the challenge is to discriminate between old and new classes without access to old data. While some methods [79, 51, 78, 80] trained the model on new classes favoring plasticity and used knowledge distillation [20] to preserve old class representations, other methods [43, 3, 13, 41, 35] froze the feature extractor after the first task, thus favoring stability and incrementally

---

[†]The corresponding author is Yuyang Liu.

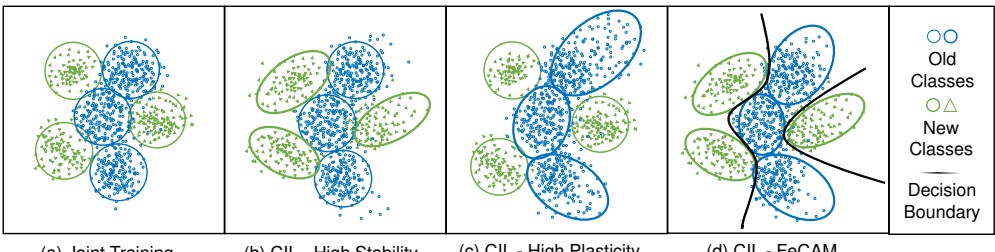

| | | | | ○○ Old Classes |
| (a) Joint Training | (b) CIL - High Stability | (c) CIL - High Plasticity | (d) CIL - FeCAM | ○△ New Classes — Decision Boundary |

Figure 1: Illustration of feature representations in CIL settings. In Joint Training (a), deep neural networks learn good isotropic spherical representations [17] and thus the Euclidean metric can be used effectively. However, it is challenging to learn isotropic representations of both old and new classes in CIL settings. When the model is too stable in (b), it is unable to learn good spherical representations of new classes and when it is too plastic in (c), it learns spherical representations of new classes but loses the spherical representations of old classes. Thus, it is suboptimal to use the isotropic euclidean distance. We propose FeCAM in (d) which models the feature covariance relations using Mahalanobis metric and learns better non-linear decision boundaries for new classes.

learned the classifier. One of the drawbacks is the inability to learn new representations with a frozen feature extractor. Inspired by transfer learning [49], the objective of these classifier-incremental methods is to make best use of the learned representations from the pretrained model and continually adapt the classifier to learn new tasks. Recently, pretrained feature extractors have been used in exemplar-free CIL by prompt-based methods [63], using linear discriminant analysis [41] and with a simple nearest class mean (NCM) classifier [22]. These methods use a transformer model pretrained on large-scale datasets like ImageNet-21k [45] and solely focus on classifier-incremental learning.

This paper investigates methods to enhance the representation of class prototypes in CIL, aiming to improve plasticity within the stability-favoring classifier-incremental setting. A standard practice in few-shot CIL [31, 42, 76, 70, 72] is to obtain the feature embeddings of new class samples and average them to generate class-wise prototypes. The test image features are then classified by computing the Euclidean distance to the mean prototypes. The Euclidean distance is used in the NCM classifier, following [17], which claims that the highly non-linear nature of learned representations eliminates the need to learn the Mahalanobis [65, 11] metric previously used [38]. Our analysis shows that this holds true for classes that are considered during training, however, for new classes, the Euclidean distance is suboptimal. To address this problem, we propose to use the anisotropic Mahalanobis distance. In Fig. 1, we explain how the feature representations vary in CIL settings. Here, the high-stability case in CIL is explored, where the model does not achieve spherical representations for new classes in the feature space, unlike joint training. Thus, it is intuitive to take into account the feature covariances while computing the distance. The covariance relations between the feature dimensions better captures the more complex class structure in the high-dimensional feature space. Additionally, in Fig. 3, we analyze singular values for old and new class features to observe the changes in variances in their feature distributions, suggesting a shift towards more anisotropic representations.

While previous methods [38] proposed learning Mahalanobis metrics, we propose using an optimal Bayes classifier by modeling the covariance relations of the features and employing class prototypes. We term this approach **Fe**ature **C**ovariance-**A**ware **M**etric (FeCAM). We compute the covariance matrix for each class from the feature embeddings corresponding to training samples and perform correlation normalization to ensure similar variances across all class representations, which is crucial for distance comparisons. We investigate various ways of using covariance relations in continual settings. We posit that utilizing a Bayes classifier enables better learning of optimal decision boundaries compared to previous attempts [67] involving feature sampling from Gaussian distributions and training linear classifiers. The proposed approach is simple to implement and requires no training since we employ a Bayes classifier. The Bayes classifier FeCAM can be used for both many-shot CIL and few-shot CIL, unlike existing methods. Additionally, we achieve superior performance with pretrained models on both class-incremental and domain-incremental benchmarks.

## 2 Related Work

Many-shot class-incremental learning (MSCIL) is the conventional setting, where sufficient training data is available for all classes. A critical aspect in many-shot CIL methods is the semantic drift in

the feature representations [68] while training on new tasks. While recent methods use knowledge distillation [20] or regularization strategies [25, 68] to maintain the representations of old classes, these methods are dependent on storing images [30, 7, 21, 14, 16, 6, 5, 24], representations [23] or instances [33] from old tasks and becomes ineffective in practical cases where data privacy is required. Another set of methods [43, 3, 13, 41, 35] proposed freezing the feature extractor after the first task and learning only the classifier on new classes. We follow the same setting and do not violate privacy concerns by storing exemplars.

On the other hand, Few-Shot Class-Incremental Learning (FSCIL) considers that very few (1 or 5) samples per class are available for training [31, 53, 58]. Generally, these settings assume a big first task and freezes the network after training on the initial classes. To address the challenges of FSCIL, various techniques have been proposed. One approach is to use meta-learning to learn how to learn new tasks from few examples [39, 48, 4, 54]. Another approach is to incorporate variational inference to learn a distribution over models that can adapt to new tasks [40]. Most FSCIL methods [1, 8, 31, 53, 42, 76, 70, 72] obtains feature embeddings of a small number of examples and averages them to get the class-wise prototypes. These methods uses the Euclidean distance to classify the features assuming equally spread classes in the feature space. We explore using the class prototypes in both MSCIL and FSCIL settings.

Since the emergence of deep neural networks, Euclidean distance is used in NCM classifier following [17], instead of Mahalanobis distance [38]. Mahalanobis distance has been recently explored for out-of-distribution detection with generative classifiers [29] and an ensemble using Mahalanobis [24] and also within the context of cross-domain continual learning [50].

# 3 Proposed Approach

## 3.1 Motivation

For the classification of hand-crafted features, Mensink *et al.* [38] proposed the nearest class mean (NCM) classifier using (squared) Mahalanobis distance $\mathcal{D}_M$ instead of Euclidean distance to assign an image to the class with the closest mean (see also illustration in Fig. 2) :

$$y^* = \underset{y=1,...,Y}{\operatorname{argmin}} \mathcal{D}_M(x, \mu_y), \quad \mathcal{D}_M(x, \mu_y) = (x - \mu_y)^T M (x - \mu_y) \tag{1}$$

where $Y$ is the number of classes, $x, \mu_y \in \mathbb{R}^D$, class mean $\mu_y = \frac{1}{|X_y|} \sum_{x \in X_y} x$, and $M$ is a positive definite matrix. They learned a low-rank matrix $M = W^T W$ where $W \in \mathbb{R}^{m \times D}$, with $m \leq D$.

However, with the shift towards deep feature representations, Guerriero *et al.* [17] assert that the learned representations with a deep convolutional network $\phi : \mathcal{X} \to \mathbb{R}^D$, eliminate the need of learning the Mahalanobis metric $M$ and the isotropic Euclidean distance $\mathcal{D}_e$ can be used as follows:

$$y^* = \underset{y=1,...,Y}{\operatorname{argmin}} \mathcal{D}_e(\phi(x), \mu_y), \quad \mathcal{D}_e(\phi(x), \mu_y) = (\phi(x) - \mu_y)^T (\phi(x) - \mu_y) \tag{2}$$

where $\phi(x), \mu_y \in \mathbb{R}^D$, $\mu_y = \frac{1}{|X_y|} \sum_{x \in X_y} \phi(x)$ is the class prototype or the average feature vector of all samples of class $y$. Here, $\phi(x)$ is the feature vector corresponding to image $x$ and could be the output of penultimate layer of the network. In Euclidean space, $M = I$, where $I$ is an identity matrix.

The success of the NCM classifier for deep learned representation (as observed by [17]) has also been adopted by the incremental learning community. The NCM classifier with Euclidean distance is now commonly used in incremental learning [44, 68, 10, 79, 72, 42, 70, 76]. However, in incremental learning, we do not jointly learn the features of all data, but are learning on a non-static data stream. As a result, the underlying learning dynamics which result in representations on which Euclidean distances perform excellently might no longer be valid. Therefore, we perform a simple comparison with the Euclidean and Mahalanobis distance (see Fig. 4) where we use a network trained on 50% of

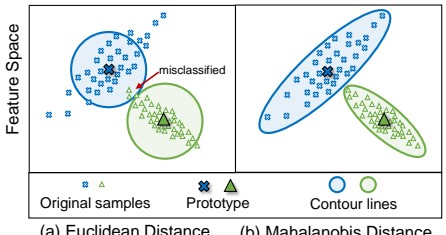

(a) Euclidean Distance    (b) Mahalanobis Distance

Figure 2: Illustration of distances (contour lines indicate points at equal distance from prototype).

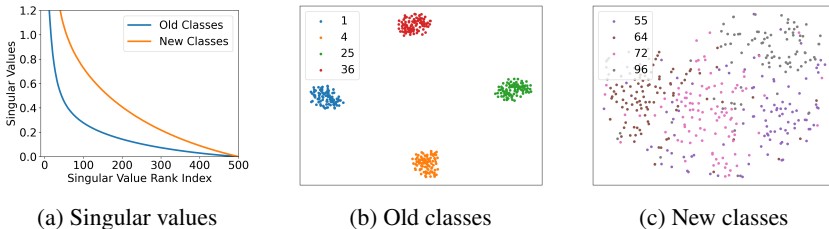

| (a) Singular values | (b) Old classes | (c) New classes |

Figure 3: (a) Singular values comparison for old and new classes, (b-c) Visualization of features for old classes and new classes by t-SNE, where the colors of points indicate the corresponding classes.

classes of CIFAR100 (identified as *old classes*). Interestingly, indeed the *old classes*, on which the feature representation is learned, are very well classified with Euclidean distance, however, for the *new classes* this no longer holds, and the Mahalanobis distance obtains far superior results.

As a second experiment, we compare singular values of *old* and *new classes* (see Fig. 3a). We observe that the singular values of new class features vary more and are in general larger than those of old classes. This points out that new class distributions are more heterogeneous (and are more widely spread) compared to the old classes and hence the importance of a heterogeneous distance measure (like Mahalanobis). This is also confirmed by a t-SNE plot of both old and new classes (see Fig. 3b,c) showing that the new classes, which were not considered while training the backbone are badly modeled by a spherical distribution assumption, as is underlying the Euclidean distance.

Based on these results, two extreme cases of CIL are illustrated in Fig. 1, considering maximum stability where the backbone is frozen, and maximum plasticity where the training is done with fine-tuning without preventing forgetting. In this paper, we revisit the nearest class mean classifier based on a heterogeneous distance measure. We perform this for classifier incremental learning, where the backbone is frozen after the first task. However, we think that conclusions based on the heterogeneous nature of class distributions also have consequences for continual deep learning where the representations are continually updated.

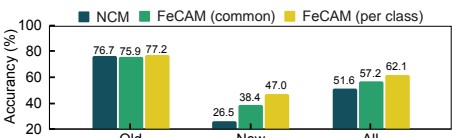

Figure 4: Accuracy Comparison of NCM (Euclidean) and FeCAM (Mahalanobis) using common covariance matrix and a matrix per class on CIFAR100 50-50 (2 task) sequence, for Old, New, and All classes at the end of the learning sequence.

### 3.2 Bayesian FeCAM Classifier

When modeling the feature distribution of classes with a multivariate normal feature distribution $\mathcal{N}(\mu_y, \Sigma_y)$, the probability of a sample feature $x$ belonging to class $y$ can be expressed as,

$$P(x|C = y) \approx \exp \frac{-1}{2}(x - \mu_y)^T \Sigma_y^{-1}(x - \mu_y), \tag{3}$$

It is straightforward to see that this is the optimal Bayesian classifier, since:

$$\operatorname*{argmax}_{y} P(Y|X) = \operatorname*{argmax}_{y} \frac{P(X|Y)P(Y)}{P(X)} = \operatorname*{argmax}_{y} P(X|Y)P(Y) = \operatorname*{argmax}_{y} P(X|Y) \tag{4}$$

where optimal boundary occurs at those points where each class is equally probable $P(y_i) = P(y_j)$.

Since logarithm is a concave function and thus $\operatorname*{argmax}_{y} P(X|Y) = \operatorname*{argmax}_{y} log P(X|Y)$

$$\operatorname*{argmax}_{y} log P(X|Y) = \operatorname*{argmax}_{y} \{-(x - \mu_y)^T \Sigma_y^{-1}(x - \mu_y)\} = \operatorname*{argmin}_{y} \mathcal{D}_M(x, \mu_y) \tag{5}$$

where the squared mahalanobis distance $\mathcal{D}_M(x, \mu_y) = (x - \mu_y)^T \Sigma_y^{-1}(x - \mu_y)$.

In the following, we elaborate on several techniques that can be applied to improve and stabilize Mahalanobis-based distance classification. We apply these, resulting in our FeCAM classifier (an ablation study in section 4.2.4 confirms the importance of these on overall performance).

**Covariance Matrix Approximation.** We obtain the covariance matrix from the feature vectors $\phi(x)$ corresponding to the samples $x$ at any task. The covariance matrix can be obtained in different ways. A common covariance matrix $\mathbf{\Sigma}^{1:t}$ can be incrementally updated as the mean covariance matrix for all seen classes till task $t$ as follows:

$$\mathbf{\Sigma}^{1:t} = \mathbf{\Sigma}^{1:t-1} \cdot \frac{|Y^{1:t-1}|}{|Y^{1:t}|} + \mathbf{\Sigma}^{t} \cdot \frac{|Y^{1:t}| - |Y^{1:t-1}|}{|Y^{1:t}|} \tag{6}$$

where $\mathbf{\Sigma}^{t}$ refers to the common covariance matrix obtained from the features of samples $x \in X^t$ from all classes seen in task $t$ and $|.|$ denotes the number of classes seen till that task. This common covariance matrix will represent the feature distribution for all seen classes and requires storing only the common covariance matrix from the last task.

The other alternative is to use a covariance matrix $\mathbf{\Sigma}_y$ for $y \in 1,..Y$ to represent the feature distribution of each class separately. Here, $\mathbf{\Sigma}_y$ is the covariance matrix obtained from the feature vectors of all the samples $x \in X_y$. This will involve storing a separate matrix for all seen classes.

**Normalization of Covariance Matrices.** The covariance matrix $\mathbf{\Sigma}_y$ obtained for each class will have different levels of scaling and variances along different dimensions. Particularly, due to the notable shift in feature distributions between the old and new classes, the variances are much higher for the new classes. As a result, the Mahalanobis distance of features from different classes will have different scaling factors, and the distances will not be comparable. So, in order to be able to use a covariance matrix per class, we perform a correlation matrix normalization on all the covariance matrices. In order to make the multiple covariance matrices comparable, we make their diagonal elements equal to 1. A normalized covariance matrix can be obtained as:

$$\hat{\mathbf{\Sigma}}_y(i,j) = \frac{\mathbf{\Sigma}_y(i,j)}{\sigma_y(i)\sigma_y(j)}, \quad \sigma_y(i) = \sqrt{\mathbf{\Sigma}_y(i,i)}, \quad \sigma_y(j) = \sqrt{\mathbf{\Sigma}_y(j,j)} \tag{7}$$

where $\sigma_y(i)$ and $\sigma_y(j)$ refers to the standard deviations along the dimensions $i$ and $j$ respectively.

We identify the difficulties of obtaining an invertible covariance matrix in cases when the number of samples are less than the number of dimensions. So, we use a covariance shrinkage method to get a full-rank matrix. We also show that a simple gaussianization of the features using tukey's normalization [55] is also helpful.

**Covariance Shrinkage.** When the number of samples available for a class is less than the number of feature dimensions, the covariance matrix $\mathbf{\Sigma}$ is not invertible, and thus it is not possible to use $\mathbf{\Sigma}$ to get the Mahalanobis distance. This is a very serious problem since most of the deep learning networks have large number of feature dimensions [69]. Similar to [27, 57], we perform a covariance shrinkage to obtain a full-rank matrix as follows:

$$\mathbf{\Sigma}_s = \mathbf{\Sigma} + \gamma_1 V_1 I + \gamma_2 V_2 (1 - I), \tag{8}$$

where $V_1$ is the average diagonal variance, $V_2$ is the average off-diagonal covariance of $\mathbf{\Sigma}$ and $I$ is an identity matrix.

**Tukey's Ladder of Powers Transformation.** Tukey's Ladder of Powers transformation aims to reduce the skewness of distributions and make them more Gaussian-like. We transform the feature vectors $\phi(x)$ using Tukey's Ladder of Powers transformation [55]. It can be formulated as:

$$\tilde{\phi(x)} = \begin{cases} \phi(x)^\lambda & \text{if } \lambda \neq 0 \\ \log(\phi(x)) & \text{if } \lambda = 0 \end{cases} \tag{9}$$

where $\lambda$ is a hyperparameter to decide the degree of transformation of the distribution. In our experiments, we use $\lambda = 0.5$ following [67].

We obtain the normalized features using Tukey's transformation and then use the transformed features to obtain the covariance matrices. When using multiple matrices, we perform correlation normalization to make them comparable. The final prediction and the squared Mahalanobis distance to the different class prototypes using one covariance matrix per class can be obtained as:

$$y^* = \underset{y=1,...,Y}{\operatorname{argmin}} \mathcal{D}_M(\phi(x), \mu_y), \quad \mathcal{D}_M(\phi(x), \mu_y) = (\tilde{\phi(x)} - \tilde{\mu}_y)^T (\hat{\mathbf{\Sigma}}_y)_s^{-1} (\tilde{\phi(x)} - \tilde{\mu}_y) \tag{10}$$

where $\tilde{\phi(x)}$ and $\tilde{\mu}_y$ refers to the tukey transformed features and prototypes respectively and $(\hat{\mathbf{\Sigma}}_\mathbf{y})_s^{-1}$ denotes the inverse of the covariance matrices which first undergoes shrinkage followed by normalization. Note that the covariance matrices are computed using the tukey transformed features. Similarly, the common covariance matrix $\mathbf{\Sigma}^{1:t}$ can be used in Eq. (10).

### 3.3 On the Suboptimality of Learning Linear Classifier

Previous methods like [67, 27] in few-shot learning assumed Gaussian distributions of classes in feature space and proposed to transfer statistics of old classes to obtain calibrated distributions of new classes and then sample examples from the calibrated distribution to train a linear logistic regression classifier. In our setting, we consider a similar baseline for comparison which assumes Gaussian distributions for features of old classes (by storing the mean and the covariance matrix from the features of the old classes) and samples features from these distributions to learn a linear classifier. We state that this is not an ideal solution since the optimal decision boundaries need not be

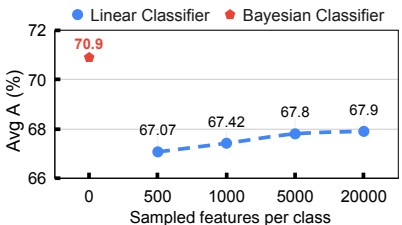

Figure 5: Avg $\mathcal{A}$cc comparison of bayesian and linear classifier on CIFAR100 (T=5) setting.

linear. The optimal decision boundaries are linear when the covariances of all classes are equal like in the Euclidean space. When the covariances of classes are not equal, the optimal decision boundaries are non-linear and forms a quadratic surface in high-dimensional feature space. We show in Fig. 5 that using the optimal Bayesian classifier obtains much better performance compared to sampling features and training a linear classifier, even when sampling many examples per class.

## 4 Experiments

### 4.1 Experimental Setup

We evaluated FeCAM with strong baselines on multiple datasets and different scenarios.

**MSCIL datasets and setup.** We conduct experiments on three publicly available datasets: 1) CIFAR100 [26] - consisting of 100 classes, 32×32 pixel images with 500 and 100 images per class for training and testing, respectively; 2) TinyImageNet [28] - a subset of ImageNet with 200 classes, 64×64 pixel images, and 500 and 50 images per class for training and testing, respectively; 3) ImageNet-Subset [12] - a subset of the ImageNet LSVRC dataset [47] consisting of 100 classes with 1300 and 50 images per class for training and testing, respectively. We divide these datasets into incremental settings, where the number of initial classes in the first task is larger and the remaining classes are evenly distributed among the incremental tasks. We experiment with three different incremental settings for CIFAR100 and ImageNet-Subset: 1) 50 initial classes and 5 incremental learning (IL) tasks of 10 classes; 2) 50 initial classes and 10 IL tasks of 5 classes; 3) 40 initial classes and 20 IL tasks of 3 classes. For TinyImageNet, we use 100 initial classes and distribute the remaining classes into three incremental settings: 1) 5 IL tasks of 20 classes; 2) 10 IL tasks of 10 classes; 3) 20 IL tasks of 5 classes. At test time, task IDs are not available.

**FSCIL datasets and setup.** We conduct experiments on three publicly available datasets: 1) CIFAR100 (described above); 2) miniImageNet [58] - consisting of 100 classes, 84×84 pixel images with 500 and 100 images per class for training and testing, respectively; 3) Caltech-UCSD Birds-200-2011 (CUB200) [59] - consisting of 200 classes, 224×224 pixel images with 5994 and 5794 images for training and testing, respectively. For CIFAR100 and miniImageNet, we divide the 100 classes into 60 base classes and 40 new classes. The new classes are formulated into 8-step 5-way 5-shot incremental tasks. For CUB200, we divide the 200 classes into 100 base classes and 100 new classes. The new classes are formulated into 10-step 10-way 5-shot incremental tasks.

**Compared methods.** We compare with several exemplar-free CIL methods in the many-shot setting [25, 44, 3, 21, 32, 68, 79, 78, 80, 43] and in the few-shot setting [58, 53, 70, 76, 8, 31, 72, 42]. Results of compared methods marked with ∗ are reproduced. For the upper bound of CIL, a joint training on all data is presented as a reference.

**Implementation details.** We use PyCIL [73] framework for our experiments. For both MSCIL and FSCIL settings, the main network architecture is ResNet-18 [18] trained on the first task using SGD with an initial learning rate of 0.1 and a weight decay of 0.0001 for 200 epochs. For the shrinkage, we use $\gamma_1 = 1$ and $\gamma_2 = 1$ for many-shot CIL and higher values $\gamma_1 = 100$ and $\gamma_2 = 100$ for few-shot CIL in our experiments. Following most methods, we store all the class prototypes. Similar to [78], we also store the covariance matrices for all classes seen until the current task. In the experiments with visual transformers, we use ViT-B/16 [15] architecture pretrained on ImageNet-

Table 1: Average top-1 incremental accuracy in exemplar-free many-shot CIL with different numbers of incremental tasks. Best results - **in bold**, second best - underlined.

| CIL Method | CIFAR-100 | | | TinyImageNet | | | ImageNet-Subset | | |
|---|---|---|---|---|---|---|---|---|---|
| | $T$=5 | $T$=10 | $T$=20 | $T$=5 | $T$=10 | $T$=20 | $T$=5 | $T$=10 | $T$=20 |
| EWC [25] | 24.5 | 21.2 | 15.9 | 18.8 | 15.8 | 12.4 | - | 20.4 | - |
| LwF-MC [44] | 45.9 | 27.4 | 20.1 | 29.1 | 23.1 | 17.4 | - | 31.2 | - |
| DeeSIL [3] | 60.0 | 50.6 | 38.1 | 49.8 | 43.9 | 34.1 | 67.9 | 60.1 | 50.5 |
| MUC [32] | 49.4 | 30.2 | 21.3 | 32.6 | 26.6 | 21.9 | - | 35.1 | - |
| SDC [68] | 56.8 | 57.0 | 58.9 | - | - | - | - | 61.2 | - |
| PASS [79] | 63.5 | 61.8 | 58.1 | 49.6 | 47.3 | 42.1 | 64.4 | 61.8 | 51.3 |
| IL2A [78] | 66.0 | 60.3 | 57.9 | 47.3 | 44.7 | 40.0 | - | - | - |
| SSRE [80] | 65.9 | 65.0 | 61.7 | 50.4 | 48.9 | 48.2 | - | 67.7 | - |
| FeTrIL* [43] | 67.6 | 66.6 | 63.5 | 55.4 | 54.3 | 53.0 | 73.1 | 71.9 | 69.1 |
| Eucl-NCM | 64.8 | 64.6 | 61.5 | 54.1 | 53.8 | 53.6 | 72.2 | 72.0 | 68.4 |
| FeCAM (ours) - $\Sigma^{1:t}$ | 68.8 | 68.6 | 67.4 | 56.0 | 55.7 | 55.5 | 75.8 | 75.6 | 73.5 |
| FeCAM (ours) - $\Sigma_y$ | **70.9** | **70.8** | **69.4** | **59.6** | **59.4** | **59.3** | **78.3** | **78.2** | **75.1** |
| Upper Bound | 79.2 | 79.2 | 79.2 | 66.1 | 66.1 | 66.1 | 84.7 | 84.7 | 84.7 |

21k [52]. The extracted features are 512 dimensional when using Resnet-18 and 768 dimensional when using pretrained ViT. More implementation details for all hyperparameters are provided in the supplementary material.

## 4.2 Experimental Results

### 4.2.1 Many-shot CIL Results

The results for an exemplar-free MSCIL setup are presented in Table 1. We present the results for a set of different MSCIL methods, joint training of a classifier, and a simple NCM classifier (Eucl-NCM) on a frozen backbone. FeCAM outperformed all others by a large margin in all settings. FeCAM version with $\Sigma^{1:t}$, storing a single covariance matrix representing all classes, already gave significantly better results than the current state-of-the-art method - FeTrIL. However, FeCAM with covariance matrix approximation $\Sigma_y$ pushes the average incremental accuracy even higher and present excellent results. It is worth noticing that Eucl-NCM outperforms many existing CIL methods. Only FeTrIL performs better than Eucl-NCM on all datasets. In Fig. 6, we present accuracy curves after each task for ten task scenarios. SSRE has a lower starting point due to a different network architecture. The rest of the methods, despite having the same starting point, end up with very different accuracies at the last task. Eucl-NCM still presents more competitive results than SSRE. FeTrIL presents better performance but is still far from FeCAM with a common covariance matrix. The FeCAM with a covariance matrix per class outperforms all other methods starting from the first incremental task. Here, in comparison to common covariance matrix, we pay the price in memory and need to store covariance matrix per class. Despite storing a matrix per class, we have less memory overhead compared to exemplar-based methods and do not violate privacy concerns by storing images.

Additionally, we compare our method against popular exemplar-based CIL methods in Table 3, where the memory buffer is set to 2K exemplars. Our method outperforms all others that do not expand the model significantly (see #P column for the number of parameters after the last task). Only Dynamically Expandable Representation (DER), which grows the model almost six times, can outperform our method.

### 4.2.2 Experiments with pre-trained models

In Table 2 different settings for exemplar-free MSCIL are presented where we follow experimental settings of Learning-to-Prompt (L2P) method [63]. Here, all methods use a ViT encoder pre-trained on ImageNet-21K. L2P [63] is a strong baseline that does not train the encoder and learns an additional 46K prompt parameters. However, as Janson *et al.* [22] presented, a simple NCM classifier can perform better for some datasets in CIL, e.g., Split-ImageNet-R [19], Split-CIFAR100 [26] and for domain-incremental learning on CoRe50 [34].

We use the widely-used benchmark in continual learning, Split-CIFAR-100 which splits the original CIFAR-100 [26] into 10 tasks with 10 classes in each task unlike the other settings in Table 1, which have different task splits. Based on ImageNet-R [19], Split-ImageNet-R was recently proposed

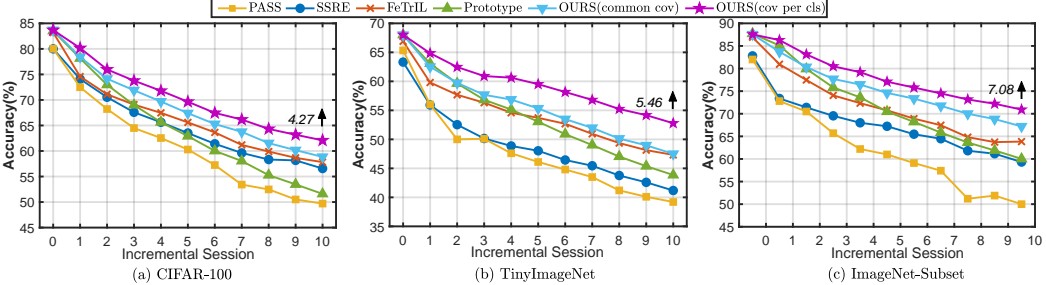

Figure 6: Accuracy of each incremental task for: (a) CIFAR100, (b) TinyImageNet, and (c) ImageNet-Subset and multiple MSCIL methods. We annotate the Avg $\mathcal{A}$cc. of all sessions between FeCAM and the runner-up method at the end of each curve. Here, prototype refers to NCM with euclidean distance.

Table 2: Avg $\mathcal{A}$cc or Test $\mathcal{A}$cc at the end of the last task on class-incremental Split-Cifar100 [26], Split-ImageNet-R [19] and domain-incremental CoRe50 [34] benchmarks. All methods are initialized with pretrained weights from ViT-B/16 [15] for fair comparison.

| CIL Method | Split-Cifar100 Avg $\mathcal{A}$cc | Split-ImageNet-R Avg $\mathcal{A}$cc | CoRe50 Test $\mathcal{A}$cc |
|---|---|---|---|
| FT-frozen | 17.7 | 39.5 | - |
| FT | 33.6 | 28.9 | - |
| EWC [25] | 47.0 | 35.0 | 74.8 |
| LwF [30] | 60.7 | 38.5 | 75.5 |
| L2P [63] | 83.8 | 61.6 | 78.3 |
| NCM [22] | 83.7 | 55.7 | 85.4 |
| FeCAM (ours) | **85.7** | **63.7** | **89.9** |
| Joint | 90.9 | 79.1 | - |

Table 3: Comparison of our method with recent exemplar-based methods which store 2000 exemplars. For fair comparison, we show the number of parameters (in millions) by #P. Results on Imagenet-Subset excerpted from [74].

| CIL Method | #P | Ex. | CIFAR-100 (T = 5) Avg. $\mathcal{A}$cc | Last $\mathcal{A}$cc | ImageNet-Subset (T = 5) Avg. $\mathcal{A}$cc | Last $\mathcal{A}$cc |
|---|---|---|---|---|---|---|
| iCaRL [44] | 11.17 | ✓ | 65.4 | 56.3 | 62.6 | 53.7 |
| PODNet [16] | 11.17 | ✓ | 67.8 | 57.6 | 73.8 | 62.9 |
| Coil [77] | 11.17 | ✓ | - | - | 59.8 | 43.4 |
| WA [71] | 11.17 | ✓ | 69.9 | 61.5 | 65.8 | 56.6 |
| BiC [64] | 11.17 | ✓ | 66.1 | 55.3 | 66.4 | 49.9 |
| FOSTER [60] | 11.17 | ✓ | 67.9 | 60.2 | 69.9 | 63.1 |
| DER [66] | 67.02 | ✓ | **73.2** | **66.2** | 77.6 | **71.1** |
| MEMO [75] | 53.14 | ✓ | - | - | 76.7 | 70.2 |
| FeCAM(ours) | 11.17 | ✗ | 70.9 | 62.1 | **78.3** | 70.9 |

by [62] for continual learning which contains 200 classes randomly divided into 10 tasks of 20 classes each. It contains data with different styles like cartoon, graffiti and origami, as well as hard examples from ImageNet with a high intra-class diversity making it more challenging for CIL experiments. We use CoRe50 [34] for domain-incremental settings where the domain of the same class of objects is changing in new tasks. It consists of 50 different types of objects from 11 domains. The first 8 domains are used for learning and the other 3 domains are used for testing. Since it has a single test task, we report the test accuracy after learning on all 8 domains similar to [63, 22]. Results of compared methods excerpted from [22].

We use the proposed FeCAM method with the pre-trained ViT using a covariance matrix per class on CIL settings. In the domain-incremental setting, we maintain a single covariance matrix per class across domains and update the matrix in every new domain by averaging the matrix from the previous domain and from the current one. FeCAM outperformed both L2P and NCM, in all the settings. Notably, FeCAM outperforms L2P by 11.55% and NCM by 4.45% on the CoRe50 dataset.

### 4.2.3 Few-shot CIL Results

In many-shot settings, it is possible to obtain a very informative covariance matrix from the large number of available samples, while it can be difficult to get a good matrix in FSCIL from just 5 samples per class. To stabilize the covariance estimation, we use higher values of $\gamma_1$ and $\gamma_2$. In Fig. 7 we present accuracy curves after each task for FSCIL on miniImageNet, CIFAR100, and CUB200 datasets. While TOPIC [53] does better than the finetuning methods, it does not perform well in comparison to the recent methods which has significantly improved the performance. ALICE [42] recently proposed to obtain compact and well-clustered features in the base task which helps in generalizing to new classes. We follow the experimental settings from ALICE [42]. We take the strong base model after the first task from ALICE and use the proposed FeCAM classifier (with a covariance matrix per class) instead of NCM classifier in the incremental tasks. We outperform ALICE significantly on all the FSCIL benchmarks.

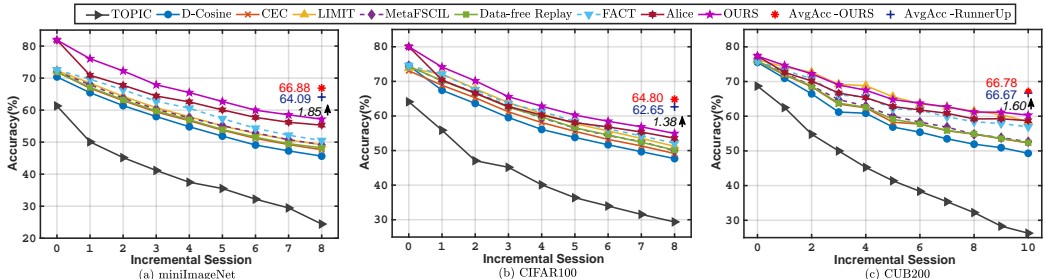

Figure 7: FSCIL methods accuracy of each incremental task for (a) miniImageNet, (b) CIFAR100 and (c) CUB200. We annotate the performance gap after the last session and Avg $\mathcal{A}$cc. of all sessions between OURS and the runner-up method at the end of each curve. Refer to supplementary material for detailed values.

Table 4: Ablation of the performance indicating the contribution from the different components of our proposed method FeCAM for MSCIL with five tasks on CIFAR-100 and ImageNet-Subset datasets. Note that here we use variance normalization (different from Eq. (7)) when using diagonal matrix.

| Distance | Cov. Matrix | Tukey Eq. (9) | Shrinkage Eq. (8) | Norm. Eq. (7) | CIFAR-100 (T=5) | | ImageNet-Subset (T=5) | |
|---|---|---|---|---|---|---|---|---|
| | | | | | Last Acc | Avg Acc | Last Acc | Avg Acc |
| Euclidean | - | ✗ | - | - | 51.6 | 64.8 | 60.0 | 72.2 |
| Euclidean | - | ✓ | - | - | 54.4 | 66.6 | 66.2 | 73.6 |
| Mahalanobis | Full | ✗ | ✗ | ✗ | 14.6 | 29.7 | 33.5 | 45.1 |
| Mahalanobis | Full | ✓ | ✗ | ✗ | 20.6 | 36.2 | 54.0 | 65.6 |
| Mahalanobis | Full | ✗ | ✓ | ✗ | 44.6 | 59.3 | 39.9 | 56.9 |
| Mahalanobis | Full | ✓ | ✓ | ✗ | 52.1 | 62.8 | 56.5 | 67.3 |
| Mahalanobis | Diagonal | ✓ | ✓ | ✗ | 55.2 | 66.9 | 64.0 | 74.1 |
| Mahalanobis | Full | ✗ | ✓ | ✓ | 55.4 | 65.9 | 58.1 | 68.5 |
| Mahalanobis | Full | ✓ | ✓ | ✓ | **62.1** | **70.9** | **70.9** | **78.3** |

#### 4.2.4   Ablation Studies

FeCAM propose to use multiple different components to counteract the CIL effect on the classifier performance. In Table 4 the ablation study for MSCIL is presented, where contribution of each component is exposed in a meaning of average incremental and last task accuracy. Here, we consider the settings using a covariance matrix per class. We show that Tukeys transformation significantly reduces the skewness of the distributions and improves the accuracy using both Euclidean and Mahalanobis distances. The effect of the covariance shrinkage is more significant for CIFAR100 which has 500 images per class (less than 512 dimensions of feature space) while it also improves the performance on ImageNet-Subset. Here, we also show the usage of the diagonal matrix where we use only the diagonal (variance) values from the covariance matrices. We divide the diagonal matrices by the norm of the diagonal to normalize the variances. When using the diagonal matrix, the storage space is reduced from $D^2$ to $D$. Finally, we show that using the correlation normalization from Eq. (7) gives the best accuracy by tackling the variance shift and better using the feature covariances.

**Time complexity.** While previous methods train the classifier [43] or the model [79, 80] for several epochs in the new tasks, we do not perform any such training in new tasks. Among the existing methods, FeTrIL [43] claims to be the fastest. We compare the time taken for the incremental tasks on ImageNet-Subset (T=5) for FeTrIL and the proposed FeCAM method. Using one Nvidia RTX 6000 GPU, FeTrIL takes 44 minutes to complete all the new tasks while FeCAM takes only 6 minutes.

**Feature transformations.** We analyze the t-SNE plot for the feature distributions of old and new classes from CIFAR100 50-50 (2 tasks) setting in different scenarios in Fig. 8. When the model is trained jointly on all classes, the features of all classes are well clustered and separated from each other. In CIL settings with frozen backbone when the model is trained on only the first 50 classes, the features are well-clustered for the old classes while the features for new classes are scattered and not well-separated. When we make the feature distributions more gaussian using Tukeys transformation, we observe that the new class features are comparatively better clustered.

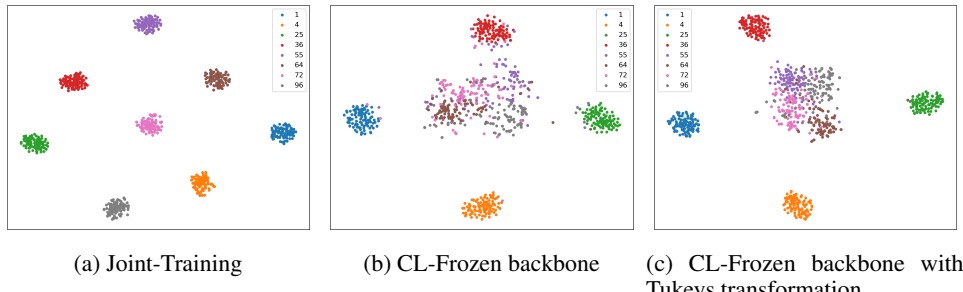

| (a) Joint-Training | (b) CL-Frozen backbone | (c) CL-Frozen backbone with Tukeys transformation |

Figure 8: The t-SNE plot for the features of new and old classes after Joint-Training (a) and after learning only the first 50 classes (b,c). In Joint-Training, the features are well clustered for all classes, however when the feature extractor is trained only on the first 50 classes, the new class representations are spread out. On applying Tukeys transformation, the new class embeddings are better clustered.

Table 5: Analysis of performance when the first task has 20 classes only and 20 new classes are added in incremental tasks, on CIFAR-100 and ImageNet-Subset datasets.

| Method | CIFAR-100 | | ImageNet-Subset | |
|---|---|---|---|---|
| | Last Acc | Avg Acc | Last Acc | Avg Acc |
| Euclidean-NCM | 30.6 | 50.0 | 35.0 | 54.5 |
| FeTrIL | 46.2 | 61.3 | 48.4 | 63.1 |
| FeCAM (ours) | **48.1** | **62.3** | **52.3** | **66.4** |

**CIL settings with a small first task.** Usually one method sticks to one setting. Exemplar-free methods use 50% of data in the first task as equally splitting is a much more challenging setting which is usually tackled by storing exemplars or by expanding the network in new tasks.

When half of the total classes is present in the first task, the feature extractor is better. When we start with fewer classes (20 classes in the first step) and add 20 new classes at every task, we can observe the same behavior in Table 5. FeCAM still works and outperforms other methods. However, the average incremental accuracy is not very high in this challenging setting because the representation learned in the first task is not as good as in the big first task setting.

## 5   Conclusions

In this paper, we revisit the anisotropic Mahalanobis distance for exemplar-free CIL and propose using it to effectively model covariance relations for classification based on prototypes. We analyze the heterogeneity in the feature distributions of the new classes that are not learned by the feature extractor. To address the feature distribution shift, we propose our Bayes classifier method FeCAM, which uses Mahalanobis distance formulation and additionally uses some techniques like correlation normalization, covariance shrinkage, and Tukey's transformation to estimate better covariance matrices for continual classifier learning. We validate FeCAM on both many- and few-shot incremental settings and outperform the state-of-the-art methods by significant margins without the need for any training steps. Additionally, FeCAM evaluated in the class- and domain-incremental benchmarks with pretrained vision transformers yields state-of-the-art results. FeCAM does not store any exemplars and performs better than most exemplar-based methods on many-shot CIL settings. As a future work, FeCAM can be adapted to CIL settings where the feature representations are continually learned.

**Limitations.** The proposed approach needs a strong feature extractor or a large amount of data in the first task to learn good representations, as we do not learn new features but reuse the ones learned on the first task (or from pretrained network). Therefore, the method is not apt when training from scratch, starting with small tasks. We would then need to extend the theory to feature distributions which undergo feature drift during training; next to prototype drift [68] also covariance changes should be modeled.

**Acknowledgement.** We acknowledge projects TED2021-132513B-I00 and PID2022-143257NB-I00, financed by MCIN/AEI/10.13039/501100011033 and FSE+ and the Generalitat de Catalunya CERCA Program. Bartłomiej Twardowski acknowledges the grant RYC2021-032765-I.

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
