# Supplementary Materials for FeCAM: Exploiting the Heterogeneity of Class Distributions in Exemplar-Free Continual Learning

**Dipam Goswami[1,2]**   **Yuyang Liu[3,4,5]**   **Bartłomiej Twardowski [1,2,6]**   **Joost van de Weijer[1,2]**
[1]Department of Computer Science, Universitat Autònoma de Barcelona
[2]Computer Vision Center, Barcelona  [3]University of Chinese Academy of Sciences
[4]State Key Laboratory of Robotics, Shenyang Institute of Automation, Chinese Academy of Sciences
[5]Institutes for Robotics and Intelligent Manufacturing, Chinese Academy of Sciences  [6]IDEAS-NCBR
{dgoswami, btwardowski, joost}@cvc.uab.es, sunshineliuyuyang@gmail.com

## 1   Definitions

The Mahalanobis distance is generally used to measure the distance between a data sample $x$ and a distribution $\mathcal{D}$. Given the distribution has a mean representation $\mu$ and an invertible covariance matrix $\boldsymbol{\Sigma} \in \mathbb{R}^{D \times D}$, then the squared Mahalanobis distance can be expressed as:

$$\mathcal{D}_M(x, \mu) = (x - \mu)^T \boldsymbol{\Sigma}^{-1} (x - \mu) \tag{1}$$

where $\boldsymbol{\Sigma}^{-1}$ is the inverse of the covariance matrix.

The covariance matrix is symmetric in nature and can be defined as:

$$\boldsymbol{\Sigma}(i, j) = \left\{ \begin{array}{ll} var(i) & i = j \\ cov(i, j) & i \neq j \end{array} \right. \tag{2}$$

where $i, j \in 1, ...D$, $var(i)$ denotes the variance of the data along the $ith$ dimension and $cov(i, j)$ denotes the covariance between the dimensions $i$ and $j$. The diagonals of the matrix represent the variances and the non-diagonal entries are the covariance values.

In euclidean space, $\boldsymbol{\Sigma} = I$, where $I$ is an identity matrix. Thus, in euclidean space, we consider identical variance along all dimensions and ignore the positive and negative correlations between the variables.

## 2   Implementation Details

We analyze the effect of the covariance shrinkage hyperparamaters $\gamma_1$ and $\gamma_2$ in Fig. 1 for the many-shot setting (T=5) on Cifar100. Based on the observations, we see that the chosen parameters $\gamma_1 = 1$ and $\gamma_2 = 1$ obtain good results. Similarly, we use $\gamma_1 = 1$ and $\gamma_2 = 1$ for all many-shot experiments on CIFAR100, TinyImageNet and ImageNet-Subset. We use $\gamma_1 = 1$ and $\gamma_2 = 0$ for the experiments on Split-CIFAR100 and Core50 datasets. For Split-ImageNet-R, We use $\gamma_1 = 10$ and $\gamma_2 = 10$. For all the few-shot CIL settings, we obtain better results with $\gamma_1 = 100$ and $\gamma_2 = 100$.

Since the Resnet-18 feature extractor uses a ReLU activation function, the feature representation values are all non-negative, so the inputs to tukey's ladder of powers transformation are all valid. However, when using the ViT encoder pre-trained on ImageNet-21K, we also have negative values in the feature representations, hence we do not apply the tukey's transformation on the features for those experiments.

**Evaluation.** Similar to [5, 11, 10], we evaluate the methods in terms of average incremental accuracy. Average incremental accuracy $A_{inc}$ is the average of the accuracy $a_t$ of all incremental tasks (including

37th Conference on Neural Information Processing Systems (NeurIPS 2023).

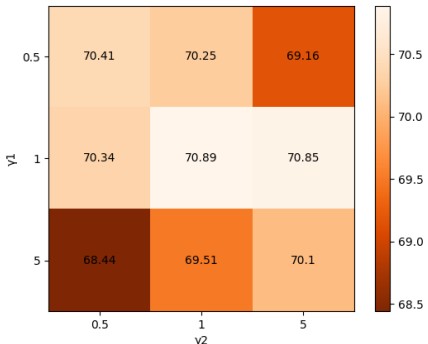

Figure 1: Impact of covariance shrinkage hyperparameters on many-shot CIFAR100 (T=5) setting using the proposed FeCAM method

the first task) and is a fair metric to compare the performances of different methods across multiple tasks.

$$A_{inc} = \frac{1}{T} \sum_{t=1}^{t=T} a_t \tag{3}$$

## 3 Further Analysis

**Storage requirements.** We analyze the storage requirements of FeCAM and compare it with the exemplar-based CIL methods in Table 1 for ImageNet-Subset (T=5) setting. Due to the symmetric nature of covariance matrices, we can store half (lower or upper triangular) of the covariance matrices and reduce the storage to half. While most of the exemplar-based methods preferred a constant storage requirement of 2000 exemplars, storage requirement for FeCAM gradually increases across steps and is still less by about 206 MBs after the last task.

Table 1: Analysis of storage requirements across tasks for FeCAM and the exemplar-based methods (storing 2000 exemplars) for the ImageNet-Subset (T=5) setting.

| Method | Task 0 | Task 1 | Task 2 | Task 3 | Task 4 | Task 5 |
|---|---|---|---|---|---|---|
| Exemplar-based | 312 MB | 312 MB | 312 MB | 312 MB | 312 MB | 312 MB |
| FeCAM (ours) | 53 MB | 63 MB | 75 MB | 85 MB | 96 MB | 106 MB |

**Pre-training with dissimilar classes.** Similar to [2], we perform experiments using the DeiT-S/16 vision transformer pretrained on the ImageNet data with different pre-training data splits and then evaluate the performance of NCM (with euclidean distance) and the proposed FeCAM method on Split-CIFAR100 (10 tasks with 10 classes in each task). In order to make sure that the pretrained classes are not similar to the classes of CIFAR100, [2] manually removed 389 classes from the 1000 classes in ImageNet. We take the publicly available DeiT-S/16 weights pre-trained on remaining 611 classes of ImageNet by [2] and evaluate NCM and FeCAM as shown in Table 2. As expected, the performance of both methods drops a bit when the pre-training is not done on the similar classes. Still FeCAM outperforms NCM by about 10% on the final accuracy. Thus, this experiment further validates the effectiveness of modeling the covariance relations using our FeCAM method in settings where images from the initial task are dissimilar to new task images.

## 4 Few-Shot CIL results

FeCAM can easily be adapted to available few-shot methods in CIL since most methods obtain class prototypes from few-shot data of new classes and then use the euclidean distance for classification.

Table 2: Performance of FeCAM and NCM-euclidean using Deit-S/16 pretrained transformer on Split-CIFAR100 dataset.

| Method | DeiT pre-trained on 1k classes | | DeiT pre-trained on 611 classes [2] | |
|---|---|---|---|---|
| | Last Acc | Avg Acc | Last Acc | Avg Acc |
| Euclidean-NCM | 60.5 | 71.4 | 58.5 | 69.2 |
| FeCAM (ours) | **70.2** | **78.5** | **68.6** | **76.9** |

We show in our paper that starting from the base task model from ALICE and simply using the FeCAM metric for classification significantly improves the performance across all tasks for the standard few-shot CIL benchmarks.

We report the average accuracy after each task for all methods on Cifar100 in Table 3, on CUB200 in Table 4 and on miniImageNet in Table 5.

Table 3: Detailed accuracy of each incremental session on CIFAR100 dataset. Best among columns in **bold**.

| Method | Accuracy in each session (%) | | | | | | | | | Avg $\mathcal{A}$ |
|---|---|---|---|---|---|---|---|---|---|---|
| | 0 | 1 | 2 | 3 | 4 | 5 | 6 | 7 | 8 | |
| Finetune | 64.10 | 39.61 | 15.37 | 9.80 | 6.67 | 3.80 | 3.70 | 3.14 | 2.65 | 16.54 |
| D-Cosine [6] | 74.55 | 67.43 | 63.63 | 59.55 | 56.11 | 53.80 | 51.68 | 49.67 | 47.68 | 58.23 |
| CEC [7] | 73.07 | 68.88 | 65.26 | 61.19 | 58.09 | 55.57 | 53.22 | 51.34 | 49.14 | 59.53 |
| LIMIT [9] | 73.81 | 72.09 | 67.87 | 63.89 | 60.70 | 57.77 | 55.67 | 53.52 | 51.23 | 61.84 |
| MetaFSCIL [1] | 74.50 | 70.10 | 66.84 | 62.77 | 59.48 | 56.52 | 54.36 | 52.56 | 49.97 | 60.79 |
| Data-free Replay [3] | 74.40 | 70.20 | 66.54 | 62.51 | 59.71 | 56.58 | 54.52 | 52.39 | 50.14 | 60.78 |
| FACT [8] | 74.60 | 72.09 | 67.56 | 63.52 | 61.38 | 58.36 | 56.28 | 54.24 | 52.10 | 62.24 |
| ALICE [4] | **80.03** | 70.38 | 66.6 | 62.72 | 60.28 | 58.06 | 56.83 | 55.35 | 53.56 | 62.65 |
| ALICE+FeCAM | **80.03** | **74.15** | **70.16** | **65.57** | **62.82** | **60.25** | **58.46** | **56.86** | **54.94** | **64.80** |

Table 4: Detailed accuracy of each incremental session on CUB200 dataset. Best among columns in **bold**.

| Method | Accuracy in each session (%) | | | | | | | | | | | Avg $\mathcal{A}$ |
|---|---|---|---|---|---|---|---|---|---|---|---|---|
| | 0 | 1 | 2 | 3 | 4 | 5 | 6 | 7 | 8 | 9 | 10 | |
| Finetune | 68.68 | 43.70 | 25.05 | 17.72 | 18.08 | 16.95 | 15.10 | 10.06 | 8.93 | 8.93 | 8.47 | 21.97 |
| D-Cosine [6] | 75.52 | 70.95 | 66.46 | 61.20 | 60.86 | 56.88 | 55.40 | 53.49 | 51.94 | 50.93 | 49.31 | 59.36 |
| CEC [7] | 75.85 | 71.94 | 68.50 | 63.50 | 62.43 | 58.27 | 57.73 | 55.81 | 54.83 | 53.52 | 52.28 | 61.33 |
| LIMIT [9] | 76.32 | 74.18 | **72.68** | 69.19 | **68.79** | 65.64 | 63.57 | 62.69 | **61.47** | 60.44 | 58.45 | 66.67 |
| MetaFSCIL [1] | 75.90 | 72.41 | 68.78 | 64.78 | 62.96 | 59.99 | 58.30 | 56.85 | 54.78 | 53.82 | 52.64 | 61.93 |
| Data-free Replay [3] | 75.90 | 72.14 | 68.64 | 63.76 | 62.58 | 59.11 | 57.82 | 55.89 | 54.92 | 53.58 | 52.39 | 61.52 |
| FACT [8] | **77.92** | 74.94 | 71.57 | 66.32 | 65.96 | 62.49 | 61.23 | 59.76 | 57.94 | 57.56 | 56.41 | 64.70 |
| FACT+FeCAM | **77.92** | **75.34** | 72.23 | 67.56 | 67.02 | 63.50 | 62.39 | 61.25 | 59.84 | 59.10 | 57.89 | 65.80 |
| ALICE [4] | 77.34 | 72.64 | 70.17 | 66.68 | 65.34 | 62.78 | 61.81 | 60.84 | 59.22 | 59.26 | 58.70 | 64.98 |
| ALICE+FeCAM | 77.34 | 74.64 | 72.22 | 69.02 | 67.50 | 64.82 | **63.74** | **62.70** | 61.20 | **61.14** | **60.30** | **66.78** |

Table 5: Detailed accuracy of each incremental session on miniImageNet dataset. Best among columns in **bold**.

| Method | Accuracy in each session (%) | | | | | | | | | Avg $\mathcal{A}$ |
|---|---|---|---|---|---|---|---|---|---|---|
| | 0 | 1 | 2 | 3 | 4 | 5 | 6 | 7 | 8 | |
| Finetune | 61.31 | 27.22 | 16.37 | 6.08 | 2.54 | 1.56 | 1.93 | 2.6 | 1.4 | 13.45 |
| D-Cosine [6] | 70.37 | 65.45 | 61.41 | 58.00 | 54.81 | 51.89 | 49.10 | 47.27 | 45.63 | 55.99 |
| CEC [7] | 72.00 | 66.83 | 62.97 | 59.43 | 56.70 | 53.73 | 51.19 | 49.24 | 47.63 | 57.75 |
| LIMIT [9] | 72.32 | 68.47 | 64.30 | 60.78 | 57.95 | 55.07 | 52.70 | 50.72 | 49.19 | 59.06 |
| MetaFSCIL [1] | 72.04 | 67.94 | 63.77 | 60.29 | 57.58 | 55.16 | 52.90 | 50.79 | 49.19 | 58.85 |
| Data-free Replay [3] | 71.84 | 67.12 | 63.21 | 59.77 | 57.01 | 53.95 | 51.55 | 49.52 | 48.21 | 58.02 |
| FACT [8] | 72.56 | 69.63 | 66.38 | 62.77 | 60.6 | 57.33 | 54.34 | 52.16 | 50.49 | 60.70 |
| ALICE [4] | **81.87** | 70.88 | 67.77 | 64.41 | 62.58 | 60.07 | 57.73 | 56.21 | 55.31 | 64.09 |
| ALICE+FeCAM | **81.87** | **76.06** | **72.24** | **67.92** | **65.49** | **62.69** | **59.98** | **58.54** | **57.16** | **66.88** |

For further analysis to demonstrate the applicability of FeCAM, we take the base task model from FACT [8] and use FeCAM in the incremental tasks for the CUB200 dataset. FeCAM improves the performance on all tasks when applied to FACT as shown in Table 4.

One of the main drawbacks of the many-shot continual learning methods is overfitting on few-shot data from new classes and hence these methods are not suited for few-shot settings. FeCAM is a

single solution for both many-shot and few-shot settings and thus can be applied in both continual learning settings.

## 5 Pseudo Code

In Algorithm 1, we present the pseudo code for using FeCAM classifier.

---
**Algorithm 1** FeCAM
---
**Require:** Training data $(D_1, D_2, .., D_T)$, Test data for evaluation $(X_1^e, X_2^e, .., X_T^e)$, Model $\phi$
1: **for** task $t \in [1, 2, .., T]$ **do**
2:     **if** $t == 1$ **then**
3:         Train $\phi$ on $D_1 = (X_1, Y_1)$                         $\triangleright$ Train the feature extractor
4:     **end if**
5:     **for** $y \in Y_t$ **do**
6:         $\mu_y = \frac{1}{|X_y|} \sum_{x \in X_y} \phi(x)$                  $\triangleright$ Compute the prototypes
7:         $\phi(\tilde{X}_y) = Tukeys(\phi(X_y))$           $\triangleright$ Tukeys transformation Eq. (9)
8:         $\mathbf{\Sigma}_y = Cov(\phi(\tilde{X}_y))$            $\triangleright$ Compute the covariance matrices
9:         $(\mathbf{\Sigma}_y)_s = Shrinkage(\mathbf{\Sigma}_y)$        $\triangleright$ Apply covariance shrinkage Eq. (8)
10:         $(\hat{\mathbf{\Sigma}}_y)_s = Normalization((\mathbf{\Sigma}_y)_s)$    $\triangleright$ Apply correlation normalization Eq. (7)
11:     **end for**
12:     **for** $x \in X_t^e$ **do**
13:         $y^* = \underset{y=1,...,Y_t}{\mathrm{argmin}} \ \mathcal{D}_M(\phi(x), \mu_y)$ where
14:         $\mathcal{D}_M(\phi(x), \mu_y) = (\phi(\tilde{x}) - \tilde{\mu}_y)^T (\hat{\mathbf{\Sigma}}_y)_s^{-1} (\phi(\tilde{x}) - \tilde{\mu}_y)$
15:                        $\triangleright$ Compute the squared mahalanobis distance to prototypes
16:     **end for**
17: **end for**
---