# OpenReview forum: "FeCAM: Exploiting the Heterogeneity of Class Distributions in Exemplar-Free Continual Learning"
_NeurIPS.cc/2023/Conference — NeurIPS 2023 poster_

### Official Review · Reviewer_QHUR · 2023-06-20

**Soundness:** 3 good
**Presentation:** 3 good
**Contribution:** 3 good
**Rating:** 6
**Confidence:** 5

**Summary:**

This paper addresses the problem of continual learning, which holds great importance in the machine learning field. The authors argue that using the Mahalanobis distance is more optimal than the Euclidean metric for handling new classes. The proposed method is evaluated on various continual learning settings and compared against state-of-the-art approaches.

**Strengths:**

1. Continual learning is of great importance in the machine learning field, and the availability of code is appreciated.
2. The proposed method is evaluated on multiple continual learning settings and compared against state-of-the-art approaches.
3. The figures effectively illustrate the distribution of old and new classes, making the proposed approach reasonable.


**Weaknesses:**

1. The largest concern about this paper is the assumption of pre-trained model (or say, relying on large number of classes in the first task). There are two typical settings of class-incremental learning, training from scratch and training from half. The former equally assigns classes into each incremental task, while the latter treats half of the total classes in the first task. According to a recent work [1], these different settings concentrate on different aspects of the continual learning algorithms. In this paper, the experiments (as well as empirical observations) are conducted on the training from half setting, making it less convincing on the other setting. Would the proposed method work for the other setting? Will the observations of covariance be the same on the former setting? More empirical evaluations and experimental comparisons are essential.
2. In figure 1, why does the training process lead to the gaussian-like distribution of the seen classes and non-gaussian-like distribution of unseen classes? Perhaps some discussions about theoretical insights behind such phenomena should be addressed.
3. The current proposed method is a combination of several mature tricks, e.g., Mahalanobis distance, Covariance Shrinkage, and Tukey’s transformation. I understand that the most contributions lie in the motivations in the preliminary part, while the combination still makes the novelty/contribution to share with former works.

[1] Online Hyperparameter Optimization for Class-Incremental Learning. AAAI 2023


**Questions:**

Remarks
1. In Figure 5, clarification is needed regarding how the authors managed to sample 20,000 instances per class for CIFAR100, as the maximum number of instances per class is 500 for CIFAR.
2. What is “classifier incremental learning” in Line 137?
3. It would be beneficial to provide an implementation guideline with pseudo code.



In summary, this paper addresses an interesting phenomenon in the field of continual learning and proposes a simple baseline method to address it. The proposed approach is evaluated on various protocols, including many-shot CIL, few-shot CIL, and pre-trained CIL, demonstrating superior performance compared to state-of-the-art methods. Although there are some evident drawbacks, the novel findings presented in this paper contribute to the continual learning field. Therefore, my initial rating is a borderline accept.

---

> ### Author Rebuttal · Authors · 2023-08-06
>
> Thanks for your feedback and suggestions.
>
> __W1:__ We agree that usually one method sticks to one setting. Exemplar-free methods use 50% of data in the first task as equally splitting is a much more challenging setting which is usually tackled by storing exemplars or by expanding the network in new tasks.
>
> When half of the total classes is present in the first task, the feature extractor is better. When we start with fewer classes (20 classes in the first step) and add 20 new classes at every task, we can observe the same behavior in the table below. FeCAM still works and outperforms other methods. However, the average incremental accuracy is not very high in this challenging setting because the representation learned in the first task is not as good as in the big first task setting.
>
> | Method |        CIFAR100         | ImageNet-Subset    |
> |   :----:    |              :----:             |          :----:                |
> |              |  Avg inc. acc (Final acc)  | Avg inc. acc (Final acc)  |
> | NCM (euclidean) |  50.0 (30.6) | 54.5 (35.0) |
> | FeTrIL | 61.3 (46.2) | 63.1 (48.4) |
> | FeCAM (ours) | __62.3 (48.1)__ | __66.4 (52.3)__ |
>
> __W2:__ As previously stated by Guerriero et al. [A], the highly non-linear nature of deep neural networks leads to isotropic spherical representations of the learned classes and hence using the euclidean distance is effective for NCM classification. However, in the class-incremental learning setting with fixed feature extractor, the unseen or new classes are not explicitly learned using the non-linear deep networks and hence the representations of these classes are anisotropic and not spherical as it used to be the case before the emergence of deep neural networks [B]. We also analyze this phenomenon in Fig. 3 (b,c) of the paper.
>
> __Q1:__ This is now clarified in the paper that we sample features for the linear classifier from the gaussian distributions of the old classes (using the prototypes and the covariance matrices for the old classes). We also updated Fig. 5 to clarify this.
>
> __Q2:__ We defined the classifier incremental learning in Lines 30-35 of introduction. It refers to the settings where the feature extractor is not updated after the first task and only the classifier is learned in new tasks.
>
> __Q3:__ We now added the pseudo code in the supplementary materials. Additionally, we will release the source code of our method in GitHub for reproduction.
>
> [A] Samantha Guerriero, et al. Deepncm: Deep nearest class mean classifiers. International Conference on Learning Representations Workshop (ICLR-W), 2018.
>
> [B] Thomas Mensink, et al. Distance-based image classification: Generalizing to new classes at near-zero cost. IEEE Transactions on Pattern Analysis and Machine Intelligence (TPAMI), 2013.

---

> > ### Comment · Reviewer_QHUR · 2023-08-11
> > **Response to the authors**
> >
> > I thank the authors for the detailed response, which addresses most of my concerns. Although I find the updated figures unavailable, the response contains most of the results. I update my rating to weak accept.
> >
> > BTW, it is still very important to report the full results given various settings (e.g., training from half and scratch), and I expect the authors to report more results on the second setting in the final version. It at least answers the capability and incapability of the proposed method in various CIL settings.

---

### Official Review · Reviewer_9pER · 2023-06-21

**Soundness:** 3 good
**Presentation:** 4 excellent
**Contribution:** 3 good
**Rating:** 7
**Confidence:** 4

**Summary:**

The authors claim that feature space of a learnt network with old classes is more scattered in newly learned classes and this prevents Euclidean distance to represent the distribution properly. Therefore, they have adopted Mahalanobis distance instead to handle the heterogenety of the newly coming class distribution. To boost up the performance, covariance normalization, covariance shrinkage, covariance matrix approximation have been adopted.

**Strengths:**

- The performance presented in this paper is state of the art. Even though bayesian classifier is usually only valid for classification problem, it is still impressive to see the advance in performance.
- The settings and amount of experiments in this paper are rich enough to support the hypothesis claimed in this paper.
- Especially, thorough ablation over the components of this method is very interesting.

**Weaknesses:**

- many existing techniques comprising the proposed method make this paper somewhat short of novelty but I still think the whole idea throughout this paper is reasonable.
- In Eq (6), I dont understand why the norm of Y is used all over the equation. Y should be the target of all samples and the norm of this does not seem to be important. I think more explanation about how this equation is derived is required here.

**Questions:**

- I agree that a bayesian classifier is well suited with incremental learning. However, when it comes to applying this incremental learning procedure to other kinds of tasks such as segmentation or generative models, will it be still effective to use it? This is not a critical comment and I just want to know the opinion of the authors.

**Limitations:**

No comments required here.

---

> ### Author Rebuttal · Authors · 2023-08-05
>
> Thanks for your comments and suggestions.
>
> __W2:__ This is not the norm. We now clarified this in the paper. We use |A| to indicate the cardinality of a set A. Here, we used | | to signify the number of classes seen till that task.
>
> __Q1:__ Yes. Whenever the classification can benefit from having richer representation by using covariance matrix, FeCAM can help with getting better classifier accuracy. However, maybe the covariance matrices should be adapted differently for different tasks.
>
> We think that FeCAM can be extended to other tasks in incremental settings which apply prototype-based classification (based on Euclidean distance). Current incremental semantic segmentation methods are mostly not prototype-based, but could be adapted to this setting. In principle, we believe that FeCAM can then be improved to obtain performance gains.
>
> We think it is harder to extend theory to incremental generative models. These models typically are not using any explicit Euclidean distance (which could be replaced by the proposed metric). In the case of pseudo-replay [A] where a generative model is used for image replay of old classes, the resulting feature distribution can form realistic anisotropic distributions. However, it should be noted that these methods require an additional generative model, and are typically reported to struggle when applied to complex image datasets (like ImageNet).
>
> [A] Shin, Hanul, et al. "Continual learning with deep generative replay." Advances in neural information processing systems 30 (2017).

---

> > ### Comment · Reviewer_9pER · 2023-08-20
> >
> > My minor concern has been resolved by the author and I maintain my decision as 'Accept'.

---

### Official Review · Reviewer_YVv6 · 2023-06-30

**Soundness:** 3 good
**Presentation:** 2 fair
**Contribution:** 2 fair
**Rating:** 4
**Confidence:** 5

**Summary:**

This manuscript introduces a Bayes classifier method FeCAM to address the feature distribution shift within the realm of Continual Incremental Learning (CIL). This method uses Mahalanobis distance formulation and additionally uses some techniques like correlation normalization, covariance shrinkage, and Tukey’s transformation to estimate better covariance matrices for continual classifier learning, thereby enhancing the efficacy of the model in addressing CIL tasks.

**Strengths:**

The manuscript presents clear and logical theoretical reasoning, making it easily understandable for readers. Moreover, the method is firmly grounded in a well-defined theoretical framework.

**Weaknesses:**

1.	There exist some mistakes, for example, on line 224, it states “20 initial classes and 10 IL steps of 5 classes”. It should be corrected to “20 initial classes and 10 IL steps of 8 classes”.
2.	From table 1, it can be observed that the results of the FeTrIL are recomputed, whereas the other results are reproduced using the original configurations of the methods. This deviation contradicts the manuscript 's description. In addition, please explain how the experimental setup in this manuscript differs from that of FeTrIL.
3.	On Page 7, the datasets “Split-ImageNet-R”, “Split-CIFAR100”, “CoRe50” and the “domain-incremental learning” need to be further explained. In particular, it is important to explain the differences between Split-CIFAR100 and the previous CIFAR100 dataset.
4.	From ablation Studies, it can be observed that the experimental results in row 7 of Table 4 outperform those in row 6. Therefore, when the diagonal matrix is combined with "Tukey", "Shrinkage" and "Norm", whether the performance will be better.


**Questions:**

1.	Page 1, ‘Recent approaches to incrementally learning the classifier by freezing the feature extractor after the first task have gained much attention.’ I think it’s necessary to explain the weakness of the method after this sentence.
2.	It can be observed that the proposed method FeCAM outperforms the existing state-of-the-art (SOTA) results on all datasets under the few-shot settings. But there should further analysis to explain the advantage of the proposed FeCAM.


**Limitations:**

The proposed method relies on a pretrained network to learn good representations because the method does not learn new features but reuse the ones learned on the first task. Therefore, when training from scratch, starting with small tasks, this method may not work well.

---

> ### Author Rebuttal · Authors · 2023-08-07
>
> Thanks for the inputs and the suggestions.
>
> __W1:__ We clarify that in the main text to “50 initial classes and 10 IL steps of 5 classes”.
>
> __W2:__ We train FeTrIL with more epochs, which results in improved accuracy in Table 1 compared to the original paper. We recompute FeTrIL as this is the state-of-the-art method with code available. The results of the other methods are taken from the FeTrIL paper which were reproduced from their original papers. We now corrected the text in the paper and clarified this in more detail.
>
> __W3:__ We added the following details of these datasets and splits in the supplementary materials.
>
> We use the widely-used benchmark in continual learning, Split-CIFAR-100 which splits the original CIFAR-100 [A] into 10 tasks with 10 classes in each task unlike the other settings in Table 1 which have different task splits. Based on ImageNet-R [B], Split-ImageNet-R was recently proposed by [C] for continual learning which contains 200 classes randomly divided into 10 tasks of 20 classes each. It contains data with different styles like cartoon, graffiti and origami, as well as hard examples from ImageNet with a high intra-class diversity making it more challenging for CIL experiments. We use CoRe50 [D] for domain-incremental settings where the domain of the same class of objects is changing in new tasks. It consists of 50 different types of objects from 11 domains. The first 8 domains are used for learning and the other 3 domains are used for testing. Since it has a single test task, we report the test accuracy after learning on all 8 domains.
>
> __W4:__ As mentioned in lines 307 to 309, the diagonal matrix is normalized (differently from Eq. 7) by dividing with the norm of the diagonal. Eq. 7 does not make sense with the diagonal matrix since it does not have the covariance values and making the diagonal values one, will lose all the information. So, we perform normalization of only the variance values. The results in row 7 of Table 4 uses ‘Tukey’, ‘Covariance Shrinkage’ and ‘Normalization’ but not the normalization from Eq. 7. We now clarified this in Table 4.
>
> __Q1:__ We add the following statement in line 33: One of the drawbacks is the inability to learn new representations with a frozen feature extractor.
>
> __Q2:__ We added more analysis to explain the advantage of FeCAM, particularly in the few-shot settings:
>
> FeCAM can easily be adapted to available few-shot methods in CIL since most methods obtain class prototypes from few-shot data of new classes and then use the euclidean distance for classification. We show in our paper that starting from the base task model from ALICE [E] and simply using the FeCAM metric for classification significantly improves the performance across all tasks for the standard few-shot CIL benchmarks.
>
> For further analysis to demonstrate the applicability of FeCAM, we take the base task model from FACT [F] and use FeCAM in the incremental tasks on the CUB200 dataset. FeCAM improves the performance on all tasks when applied to FACT as shown in the table below.
>
> | Method | Task 0 | Task 1 | Task 2 | Task 3 | Task 4 | Task 5 | Task 6 | Task 7 | Task 8 | Task 9 | Task 10 | Avg |
> | :-----: | :-----: | :-----: | :-----: | :-----: | :-----: | :-----: | :-----: | :-----: | :-----: | :-----: | :-----: | :-----: |
> | FACT | __77.9__ | 74.9 | 71.6 | 66.3 | 65.9 | 62.5 | 61.2 | 59.8 | 57.9 | 57.6 | 56.4 | 64.7 |
> | FACT+FeCAM | __77.9__ | __75.3__ | __72.2__ | __67.6__ | __67.0__ | __63.5__| __62.4__ | __61.3__ | __59.8__ | __59.1__ | __57.9__ | __65.8__ |
>
> One of the main drawbacks of the many-shot continual learning methods is overfitting on few-shot data from new classes and hence these methods are not suited for few-shot settings. FeCAM is a single solution for both many-shot and few-shot settings and thus can be applied in both type of continual learning settings.
>
> [A] Alex Krizhevsky, et al. Learning multiple layers of features from tiny images. 2009.
>
> [B] Dan Hendrycks, et al. The many faces of robustness: A critical analysis of out-of-distribution generalization. In Proceedings of the IEEE/CVF International Conference on Computer Vision, 2021.
>
> [C] Wang Z, et al. Dualprompt: Complementary prompting for rehearsal-free continual learning. In European Conference on Computer Vision, 2022.
>
> [D] Vincenzo Lomonaco and Davide Maltoni. Core50: a new dataset and benchmark for continuous object recognition. In Conference on Robot Learning, pages 17–26. PMLR, 2017.
>
> [E] Can Peng, et al. Few-shot class-incremental learning from an open-set perspective. In European Conference on Computer Vision (ECCV), 2022.
>
> [F] Da-Wei Zhou, et al. Forward compatible few-shot class-incremental learning. In Proceedings of the IEEE/CVF Conference on Computer Vision and Pattern Recognition, 2022.

---

### Official Review · Reviewer_nDen · 2023-07-06

**Soundness:** 3 good
**Presentation:** 3 good
**Contribution:** 2 fair
**Rating:** 6
**Confidence:** 3

**Summary:**

The authors study classifiers in the incremental learning scenario with a fixed strong pre-trained feature extractor. They point out the limitations of the Euclidean distance-based nearest class mean classifier in continual learning and demonstrate the benefits of Mahalanobis distance-based classifier. Several methods (shrinkage, Tukey’s transformation, covariance normalization) are proposed to estimate the covariance matrices and their effectiveness is examined through ablation studies. The authors evaluate their method in various settings, including many-shot CIL, few-shot CIL, and domain incremental learning. Although the proposed method does not require saving previous samples, it outperforms the existing CL methods including those based on replay, in terms of performance.

**Strengths:**

1. The proposed method is simply, yet outperforms the existing methods
2. The proposed method does not require saving previous samples
3. The paper is well-written and easy to follow

**Weaknesses:**

1. My main concern regarding this method is that it relies on a fixed feature extractor after the initial training, with only the classifier being adapted. As [1] shows, fixed models underperform when the pre-training data and the continual learning data are dissimilar because they cannot acquire new knowledge. It’d make the paper stronger if the authors show how the model performs with a feature extractor pre-trained with dissimilar classes from CL. For instance, [2] uses a model pre-trained with ImageNet after removing the classes similar to CIFAR and Tiny-ImageNet.
2. Saving the covariance matrix for each class in the feature-level seems to be expensive for memory, especially when there are a large number of classes.

[1] Ostapenko et al., Continual learning with foundation models: An empirical study of latent replay. CoLLAs, 2022

[2] Kim et al. A multi-head model for continual learning via out-of-distribution replay. CoLLAs, 2022


**Questions:**

1. How does the performance change when using models pre-trained using classes dissimilar to the classes used in CL?
2. Can the authors provide any suggestions on how to apply their method for a trainable feature extractor?

**Limitations:**

Refer to Weaknesses and Questions.

---

> ### Author Rebuttal · Authors · 2023-08-06
>
> Thanks for the inputs and suggestions.
>
> __W1, Q1:__ Similar to [2], we perform experiments using the DeiT-S/16 vision transformer pretrained on the ImageNet data with different pre-training data splits and then evaluate the performance of NCM (with euclidean distance) and the proposed FeCAM method on Split-CIFAR100 (10 tasks with 10 classes in each task). In order to make sure that the pretrained classes are not similar to the classes of CIFAR100, [2] manually removed 389 classes from the 1000 classes in ImageNet. We take the publicly available DeiT-S/16 weights pre-trained on remaining 611 classes of ImageNet by [2] and evaluate NCM and FeCAM. As expected, the performance of both methods drops a bit when the pre-training is not done on the similar classes. Still FeCAM outperforms NCM by about 10% on the final accuracy. Thus, this experiment further validates the effectiveness of modeling the covariance relations using our FeCAM method in settings where images from the initial task are dissimilar to new task images.
>
> | Split-CIFAR100 | DeiT-S/16 pre-trained on 1k classes | DeiT-S/16 pre-trained on 611 classes [2] |
> |  :-------: | :-------: | :-----: |
> |    |  Avg Inc Acc (Final Acc) | Avg Inc Acc (Final Acc) |
> | NCM (euclidean) | 71.4 (60.5) | 69.2 (58.5) |
> | FeCAM (ours) | __78.5 (70.2)__ | __76.9 (68.6)__ |
>
> __W2:__ We want to point out that we describe this in the supplementary materials and we discuss it in comparison to exemplars-based methods where we show that FeCAM requires much less storage space as compared to exemplar-based methods.
>
> Furthermore, due to the symmetric nature of covariance matrices, we can store half (lower or upper triangular) of the covariance matrices and reduce the storage to half. The analysis of storage requirements after every task for FeCAM and the exemplar-based methods (storing 2000 exempars) for the ImageNet-Subset are as follows:
>
> | Method | Task 0 | Task 1 | Task 2 | Task 3 | Task 4 | Task 5 |
> | :----: | :----: | :----: | :----: | :----: | :----: | :----: |
> | Exemplar-based | 312 MB | 312 MB | 312 MB | 312 MB | 312 MB | 312 MB |
> | FeCAM (ours) | 53 MB | 63 MB | 74 MB | 84 MB | 95 MB | 105 MB |
>
> As future work, using covariance matrix factorization to reduce the storage requirements can be explored.
>
> SVD(covariance matrix) =  U\*S\*V
>
> Here, the singular values S are in descending order. So, one can consider taking the first k singular values from S, and similarly truncating the U and V matrices (by taking the first k columns from U and the first k rows from V). This direction can be explored since it reduces the matrix dimensions from 512\*512 to (512\*k + k + k\*512) where k should at least be less than 256.
>
> __Q2__: As a consequence of backbone feature drift, the distribution of previously seen classes change and would need to be adapted. Semantic Drift Compensation (SDC) [A] proposes a method to estimate the drift of a single point in the latent space, based on the observed drift of the current data. In [A] this method is used to update the mean of the distribution (it is then combined with the Euclidean distance). We would also need to update the covariance matrix of the distribution. A potential approach could sample a set of points from the previous distribution, apply SDC to each of them, and then compute the covariance matrix on these points. We can then use FeCAM with the updated covariance matrix and the updated prototypes.
>
> [A] Lu Yu, et al. Semantic drift compensation for class-incremental learning. In Conference on Computer Vision and Pattern Recognition (CVPR), 2020.

---

> > ### Comment · Reviewer_nDen · 2023-08-16
> > **Response to the author comments**
> >
> > Thank you for the detailed responses. My concerns are addressed. I raised my score from 5 to 6. Please reflect the discussions and experiments in the revision.

---

### Official Review · Reviewer_MMgd · 2023-07-12

**Soundness:** 3 good
**Presentation:** 3 good
**Contribution:** 3 good
**Rating:** 7
**Confidence:** 4

**Summary:**

The paper presents a method for class-incremental learning when a strong base classifier is available and incrementally classes are added to the classifier. The base classifier is kept frozen and another classifier is used on top of the features from this classifier. Instead of using deep networks, authors propose using a bayes classifier and modeling the distance between classes using mahalanobis distance instead of the popular euclidean distance. Authors argue and show that the features of the incoming new classes are heterogenous and are better suited to be measured with mahalanobis distance. The authors also propose some solutions to the problems with the covariance matrix in mahalanobis distance calculation and to improve the performance further. The method is evaluated on many and few-shot settings and evaluated on many standard datasets where the method seems to beat many existing methods.

**Strengths:**

1. The paper is very well-written and easy to follow, supplementary material also provides good background and completes the picture,
2. Authors do a good job in motivating and validating their motivation and methodology, with plots, figures and qualitative numbers. The use of mahalanobis distance and corresponding proposals to the covariance matrix seem like simple but powerful changes.
3. The method works without additional training of classifiers so it would be quicker to incorporate new classes in an incremental setting
4. It does not use significant additional storage space and the quantitatively the methods does well compared to existing methods


**Weaknesses:**

1. There are no error bars added for the results which would provide more confidence in the results, it is a convention in incremental learning research to only have a single plot per method without any error bars, earlier since it used to be computationally expensive but no reason we should start doing it now,

2. The limitations already address this, but a follow-up on changing features would exacerbate the drift problem and would love to see the theory handle that case as well, the authors already list as future work so no action is needed for this submission,


**Questions:**

1. How is the order of the classes which are added decided?
2. Which classes are selected as part of MiniImagenet and ImagenetSubset, the classes and few-shot images should also be provided for full reproducibility


**Limitations:**

Authors have sufficiently addressed the limitations of the method. The limitations make the method only applicable in certain cases but the authors acknowledge it and the performance of the method seems pretty good in that setting.

---

> ### Author Rebuttal · Authors · 2023-08-05
>
> Thanks for all the feedback and suggestions.
>
> __W1:__ FeCAM is independent of random initialization. Since we do not train any deep neural network after the first step, the method is deterministic and thus we do not see any variation in the results on multiple runs.
>
> __W2:__ As a consequence of changing features and the backbone feature drift, the distribution of previously seen classes change and would need to be adapted. Semantic Drift Compensation (SDC) [A] proposes a method to estimate the drift of a single point in the latent space, based on the observed drift of the current data. In [A] this method is used to update the mean of the distribution (it is then combined with the Euclidean distance). We would also need to update the covariance matrix of the distribution. A potential approach could sample a set of points from the previous distribution, apply SDC to each of them, and then compute the covariance matrix on these points. We can then use FeCAM with the updated covariance matrix and the updated prototypes.
>
> __Q1:__ Following iCaRL [B], PyCIL [C] and several other CIL papers, we follow the same order of classes for fair comparison (iCaRL uses seed 1993). We included the sequence of class indices in supplementary material.
>
> __Q2:__ For ImageNet-Subset in many-shot settings, we use the same classes as in PyCIL [C], which is openly available on Kaggle platform under the name: `arjunashok33/imagenet-subset-for-inc-learn`. For miniImageNet in few-shot setting, we use the same set of images in every task as done previously in [D, E, F].
>
> We will release the source code of our method in GitHub for reproduction (code for one setting is already provided in the supplementary materials). In the source code, the dataset details are more clear and can be easily used for reproducing our results.
>
> [A] Lu Yu, et al. Semantic drift compensation for class-incremental learning. In Conference on Computer Vision and Pattern Recognition (CVPR), 2020.
>
> [B] Sylvestre-Alvise Rebuffi, et al. icarl: Incremental classifier and representation learning. In Conference on Computer Vision and Pattern Recognition (CVPR), 2017.
>
> [C] Da-Wei Zhou, et al. Pycil: a python toolbox for class-incremental learning. SCIENCE CHINA Information Sciences, 2023.
>
> [D] Chi Zhang, et al. Few-shot incremental learning with continually evolved classifiers. In Proceedings of the IEEE/CVF conference on computer vision and pattern recognition, 2021.
>
> [E] Can Peng, et al. Few-shot class-incremental learning from an open-set perspective. In European Conference on Computer Vision (ECCV), 2022.
>
> [F] Da-Wei Zhou, et al. Forward compatible few-shot class-incremental learning. In Proceedings of the IEEE/CVF Conference on Computer Vision and Pattern Recognition, 2022.

---

### Author Rebuttal · Authors · 2023-08-08

We thank the reviewers for their insightful comments and sincerely appreciate their efforts in providing valuable feedback. The reviewers agree that the paper is very well-written and is easy to follow (MMgd, nDen), has clear and logical theoretical reasoning (YVv6), along with a good motivation with plots and figures (MMgd, QHUR) and also appreciated the availability of code (QHUR). The reviewers appreciated the multiple experimental settings (9pER, QHUR, MMgd) and the thorough ablation experiments (9pER).

We address all the weaknesses and questions raised by all the reviewers in the respective rebuttals. We believe that most of these inputs are valid and greatly improve our paper.

We summarize the major changes we made to the paper in the rebuttal process:

1. In order to make sure that the pretrained classes are not similar to the incremental classes, we perform experiments with the base model pre-trained with only the dissimilar classes and observe that FeCAM still performs good and improves over euclidean-NCM significantly.
2. We clarified the experimental settings for many-shot methods in more detail. We also added the details of the Split-CIFAR100, Split-ImageNet-R and CoRe50 datasets.
3. We performed more experiments to show that FeCAM can be easily adapted to multiple few-shot continual learning methods with improved performance.
4. We clarified the notations in equation 6.
5. We perform experiments in settings starting with only 20 classes in the first step and show that FeCAM still outperforms the most competitive exemplar-free method FeTrIL and NCM with euclidean distance.
6. We updated Figure 5 for more clarity. We also added a pseudo code of FeCAM.

We hope that our responses to the reviewer's questions and the additional experiments will help in the review process.

Regards,

Authors

---

### Decision · Program_Chairs · 2023-09-21

**Decision:**

Accept (poster)

**Comment:**

The authors propose an algorithm to improve the performance of exemplar-free class-incremental learning (CIL). In doing so, they studied the commonly used Euclidean metric in exemplar-free CIL and showed its shortcomings in heterogeneous cases. Based on this observation, the authors opt for an adaptive metric (based on the Mahalanobis distance) and achieve improvements over prior arts consistently. This paper was quite well received by reviewers, with scores of 6, 4, 7, 6, and 7. Reviewers felt the paper was well-written,  clear, and expressed the value of the core idea. The AC agrees with the reviewers and recommends the acceptance of the work; congratulations. Please revise your work based on comments and incorporate the new results provided during the rebuttal phase.